# High Iron Exposure from the Fetal Stage to Adulthood in Mice Alters Lipid Metabolism

**DOI:** 10.3390/nu14122451

**Published:** 2022-06-13

**Authors:** Minju Kim, Yeon-hee Kim, Sohyun Min, Seung-Min Lee

**Affiliations:** Department of Food and Nutrition, BK21 FOUR Project, College of Human Ecology, Yonsei University, Seoul 03722, Korea; minjue3517@yonsei.ac.kr (M.K.); yh0227@yonsei.ac.kr (Y.-h.K.); minsh@yonsei.ac.kr (S.M.)

**Keywords:** high dietary iron, dietary iron intake, high iron exposure, maternal iron status, ferritin heavy chain, iron homeostasis, lipid metabolism

## Abstract

Iron supplementation is recommended during pregnancy and fetal growth. However, excess iron exposure may increase the risk of abnormal fetal development. We investigated the potential side effects of high iron levels in fetuses and through their adult life. C57BL/6J pregnant mice from 2 weeks of gestation and their offspring until 30 weeks were fed a control (CTRL, FeSO_4_ 0 g/1 kg) or high iron (HFe, FeSO_4_ 9.9 g/1 kg) diets. HFe group showed higher iron accumulation in the liver with increased hepcidin, reduced TfR1/2 mRNAs, and lowered ferritin heavy chain (FTH) proteins in both liver and adipose tissues despite iron loading. HFe decreased body weight, fat weight, adipocyte size, and triglyceride levels in the blood and fat, along with downregulation of lipogenesis genes, including PPARγ, C/EBPα, SREBP1c, FASN, and SCD1, and fatty acid uptake and oxidation genes, such as CD36 and PPARα. UCP2, adiponectin, and mRNA levels of antioxidant genes such as GPX4, HO-1, and NQO1 were increased in the HFe group, while total glutathione was reduced. We conclude that prolonged exposure to high iron from the fetal stage to adulthood may decrease fat accumulation by altering ferritin expression, adipocyte differentiation, and triglyceride metabolism, resulting in an alteration in normal growth.

## 1. Introduction

Pregnant women have an increased need for iron due to the maternal expansion of red blood cell mass and fetoplacental development [1]. There is a linear relationship between the iron status of the mother and pregnancy outcomes, as well as fetus/neonate development, such as birth weight, birth length, chest/head circumferences, and preterm labor [2,3]. Alwan et al. demonstrated that iron consumption from both food and supplements during pregnancy had a positive effect on the birth weight of infants in British women [4]. In addition, women with iron-deficiency anemia and low ferritin levels (<12 μg/L) exhibit a higher prevalence of adverse pregnancy outcomes, such as delayed growth of newborns [5]. The World Health Organization (WHO) and Centers for Disease Control and Prevention (CDC) have recommended the use of iron supplements during pregnancy [6] starting from the second phase of gestation because the iron requirement drops significantly as menstruation stops and steadily increases due to higher oxygen consumption until the end of the last trimester [2].

The WHO recommends 30–60 mg/day of iron supplementation for pregnant women [7]. In Korea, according to the 2020 dietary reference intakes for Koreans, the recommended daily intake of iron for pregnant women is between 10 mg/day to a maximum of 45 mg/day [8]. As per the Third National Health and Nutrition Examination Survey (NHANES III), the recommended dietary allowance of iron for non-pregnant healthy women in the United States is 18 mg/day to an upper limit of 45 mg/day [9]. For non-pregnant women with iron-deficient anemia, the CDC recommends 60–120 mg of iron supplementation [10].

When iron consumption exceeds the recommended upper limits, symptoms of iron toxicity may appear, resulting in unwanted consequences in both pregnant women and their newborns. Severe nausea or vomiting was observed in pregnant women supplemented with a high dose of ferrous sulfate (approximately 200 mg/day) [11]. Supplementation with high iron (100 mg/day) for extended periods during pregnancy and post-partum increased the birth weight of newborns with a high risk of hemoconcentration at delivery, although it lowered the rates of iron deficiency anemia and preterm birth [12]. In contrast, Thame et al. [13] proposed that pregnant women in the highest quartile (hemoglobin level > 12.5 g/dL) during the first phase of pregnancy labored infants with lower body weight and short crown heel length, and similar outcomes were observed for women in the lowest quartile (hemoglobin level < 9.5 g/dL). In a cohort study of non-anemic South Indian pregnant women, the highest iron supplementation (>39.2 mg/day) group showed an elevated risk of low birth weight [14]. Iron absorption efficiency can be affected by a high dose of iron. When young women without anemia (ferritin level <= 20 ug/L) were supplemented with 240 mg of iron for more than two days, fractional absorption of iron decreased by more than 45% [15]. Co-supplementation of iron with vitamin C (500 mg/day) increased lipid peroxidation in women in the third trimester of pregnancy [16]. In contrast, no significant association was detected between mild iron excess (serum ferritin > 400 ng/mL) in premature infants and nervous system development [17]. The effects of iron supplementation may differ depending on the affected tissues and varying iron conditions such as dosage, duration, and status [18].

High iron levels have been implicated in dysregulated metabolism in several aspects [16,19,20,21]. Excess dietary iron is related to insulin tolerance, glucose metabolism, diabetes, and its complications [22]. Iron overload in a mouse model causes high levels of glucose and lower levels of lactate and malate in the serum, indicating a modified TCA cycle, consequently affecting glucose homeostasis [19]. High plasma hepcidin and ferritin concentrations in pregnant women at 15–26 weeks of gestation (first trimester) are significantly associated with the risk of gestational diabetes mellitus [23]. In addition, Merono et al. reported that patients with iron overload showed alterations in lipid metabolism, owing to elevated levels of triglycerides and oxidized low-density lipoproteins in addition to changes in glucose metabolism [24]. Acute iron overload in mice lowered adiponectin levels in adipocytes [25]. Iron supplementation in high-fat diets led to alterations in the expression of genes involved in lipid metabolism, including *SREBP1* and *FASN* in the mouse liver [21]. In another mouse model, iron dextran injection decreased PPARα and increased thiobarbituric acid-reactive substances (TBARS) in the liver [20]. Iron dextran injection in a mouse model caused bone loss due to increased oxidative stress and proinflammatory cytokines, such as tumor necrosis factor (TNF-α) and interleukin-6 [26].

Despite all the above studies, the effects of iron supplementation during pregnancy to determine a safe dose for a healthy metabolism of the offspring are not studied in much detail. The present study aimed to investigate the effects of high-dose dietary iron in offspring mice after excess iron consumption since their fetal period. Such a study will help to understand the relationship between high doses of iron supplementation from the fetal stage and its impact on metabolism, which will demonstrate the necessity to consider accurate iron dosage and duration during iron supplementation.

## 2. Materials and Methods

### 2.1. Animal Experimental Protocol

Nine-week-old C57BL/6J pregnant mice were obtained from Doo Yeol Biotech (Seoul, Korea) at two weeks of gestation (*n* = 6). The animals were housed at 23 ± 2 °C, with a relative humidity of 55% ± 10%, under a 12 h light/dark cycle. All the mice had free access to food and water. Pregnant mice were randomly assigned to two groups, control (CTRL) and high iron dose (HFe). Mother mice of each group were fed the experimental diets outlined in Table 1 starting from 1 weeks before giving birth, and their pups were fed the same diet as the mother mice after a weaning period of 3 weeks. The pups (CTRL = 6, HFe = 9) were euthanized at week 30. Animals were monitored daily and weighed weekly. All experimental protocols were approved by the Institutional Animal Care and Use Committee of Yonsei University, South Korea (permit number 201712-676-04).

### 2.2. Biochemistry Measurement in Blood Plasma and Tissue

Triglyceride (TG) and total cholesterol (TCHO) levels in tissues were measured using ASAN SET TG-S and ASAN SET Total Cholesterol (ASAN PHARM., Seoul, Korea), respectively, according to the manufacturer’s protocol. Free fatty acid (FFA) was analyzed using an EnzyChrom Free Fatty Acid Assay Kit (Bio-Assay Systems, Hayward, CA, USA) according to the manufacturer’s instructions. Plasma was measured using a FUJIFILM DRI-CHEM 4000i machine (FujiFilm Co., Tokyo, Japan), and FUJI DRI-CHEM slides were used for TG, TCHO, and HDL-C (high density lipoprotein-cholesterol) (Fujifilm Co., Tokyo, Japan). LDL-C (low density lipoprotein-cholesterol) levels were calculated using Frieldwann’s equation: LDL-C = [TCHO − (HDL-C + (TG/5))]. Plasma ferritin was assessed by using a commercially available enzyme linked immunosorbent assay (ELISA) kit (ab157713, Abcam, Cambridge, MA, USA).

### 2.3. Oral Glucose Tolerance Test (OGTT)

Mice were fasted for 12 h, and glucose solution was orally administered at 2 g/kg body weight as previously described [27]. OGTT levels in mice were measured with Accu-Chek glucose meter (Roche Diagnostics, Indianapolis, IN, USA) via tail vein at all stages (0, 15, 30, 60, 90, and 120 min) [28]. Results for blood glucose concentrations are expressed in mg/dL blood.

### 2.4. Histological Examination

Liver and fat tissues were fixed with 10% formalin, and the sections were sliced to 5 µm thickness for each sample by microtome. Prussian blue staining was performed to detect ferric ions in tissue sections. The sections were incubated in an oven at 65 °C and deparaffinized using xylene. They were treated with alcohol, gradually from 100% to 70%, and hydrated with water. For the Perl’s Prussian Blue staining, the sample was placed in a working solution prepared by mixing equal proportions of 20% HCl and 10% potassium ferrocyanide solution and heated to 60 °C. The sections were then rinsed with distilled water several times and dehydrated using a graded series of 70–100% alcohol in xylene after washing. Deparaffinized sections were stained with hematoxylin and eosin (H&E) stain solution, and representative photomicrographs were taken using a light microscope (Eclipse Ti microscope; Nikon, Tokyo, Japan).

### 2.5. Determination of Malondialdehyde (MDA) Level in Mouse Tissues

Approximately 10 mg of frozen mouse liver and fat tissues was used to determine the MDA levels using the spectrophotometric method. Tissues were homogenized in distilled water with the addition of 2N perchloric acid and 5% butylated hydroxytoluene (BHT). Ground tissues were centrifuged at 13,000× *g* at 4 °C. The supernatant was collected and used in the MDA assay. The thiobarbituric acid (TBA) reagent was added to each vial containing the samples. The samples were incubated at 95 °C for 1 h. The absorbance of the samples was read at 532 nm for the colorimetric assay, and the obtained OD value was calculated using the MDA standard curve.

### 2.6. Measurement of Total Glutathione (tGSH) Content

tGSH levels were measured according to a previously described protocol [29,30]. Tissues were homogenized in 7.5% trichloroacetic acid (TCA) for tGSH analysis. The homogenized samples were centrifuged at 14,000× *g* at 4 °C. The tissue samples pretreated with 7.5% TCA were diluted at a ratio of 1:100 and treated with PB200 (0.2 M phosphate buffer, 0.2 M, PH 7.4), 5,5-dithio-bis-(2-nitrobenzoic acid) (DTNB), β-Nicotinamide adenine dinucleotide phosphate (β-NADPH), sample, and 20 IU mL^−1^ glutathione reductase (GR) sequentially. The absorbance of the mixed samples was measured at 412 nm wavelength for 1 min. The absorbance of the sample was replaced with that of the GSH standard solution (10, 25, 50, 75, and 100 µM) to obtain a standard curve (Appendix A), and then the concentration of tGSH was calculated.

### 2.7. Ferrozine Essay

The iron concentrations in the samples were determined by the ferrozine method according to Sheikh et al. [31]. The samples were centrifuged at 10,000× *g*, 4 °C for 5 min, and the protein concentration was measured by Bradford assay. Equal amounts of measured protein, 10 mM HCL, and iron-release in reagent (1.4 M HCl and 4.5% (*w*/*v*) KMnO_4_ in H_2_O) were added and mixed.

### 2.8. cDNA Extraction and Quantitative Real-Time PCR

TRIzol reagent (Molecular Research Center Inc., Cincinnati, OH, USA) was used to extract mRNA from mouse liver and mesenteric fat tissues, according to the manufacturer’s protocol. One microgram of extracted RNA in DEPC water was reverse transcribed to synthesize cDNA using a commercial ImProm-II Reverse Transcriptase kit (Promega, Madison, WI, USA; A3803) with a random primer mixture according to the manufacturer’s instructions. Quantitative real-time PCR was performed using the BioFACT™ 2X Real-Time PCR Master Mix (BioFACT™, Daejeon, Korea). The expression levels were normalized to that of β-actin.

### 2.9. Western Blot

Mouse tissues were ground in RIPA buffer supplemented with 1 mM sodium diphosphate decahydrate, 1 mM β-glycerophosphate, 1 mM NaF, and 1 mM Na_3_VO_4_ with protease inhibitor (1 mM phenyl methyl sulfonyl fluoride, 160 µM aprotinin, 1 mM leupeptin, 0.5 M dithiothreitol). Extracted protein samples were boiled with 5× sample buffer for 10 min and loaded in equal amounts onto sodium dodecyl sulfate-polyacrylamide gels. Antibodies mouse anti-FTH (Santa Cruz Biotechnology Inc., Dallas, TX, USA), mouse anti-Liver X receptor alpha and beta (LXR α/β) (Santa Cruz Biotechnology Inc., Dallas, TX, USA), mouse anti- SREBP1 (Santa Cruz Biotechnology Inc., Dallas, TX, USA), mouse anti-CCAAT-enhancer-binding proteins α (C/EBPα) (Santa Cruz Biotechnology Inc., Dallas, TX, USA), rabbit anti-nuclear factor erythroid 2-related factor 2 (Nrf2) (Cell Signaling Technology, Danvers, MA, USA), and mouse anti-α-tubulin (Santa Cruz Biotechnology Inc., Dallas, TX, USA), rabbit-phosphorylated AMP-activated protein kinase (pAMPK) (Cell Signaling Technology, Danvers, MA, USA), and rabbit anti-peroxisome proliferator-activated receptor gamma coactivator 1 α (PGC1α) (Santa Cruz Biotechnology Inc., Dallas, TX, USA) were used to detect the proteins. Membranes were washed with phosphate-buffered saline and incubated with peroxidase-conjugated 1:5000 dilutions of anti-mouse (Bio-Rad Laboratories, CA, USA) or anti-rabbit secondary antibodies (Millipore Corporation, Billerica, MA, USA). Protein signals were visualized using a D-Plus ^TM^ ECL Femto System (Dongin Biotech, Seoul, Korea) and photographed using an AE-9300 Ez-Capture system (ATTO, Tokyo, Japan). Protein expression levels were quantified using ImageJ software (National Institutes of Health, Bethesda, MD, USA).

### 2.10. Methylation Specific Polymerase Chain Reaction (MSP)

Genomic DNA (gDNA) was extracted from mouse liver or fat tissues using a HiGene^TM^ Genomic DNA Prep Kit (BioFACT, Korea). The gDNA samples were modified by bisulfite conversion using the Ez DNA methylation Gold Kit (ZYMO Research, Orange, CA, USA). The gDNA was denatured with heat, followed by a conversion reaction. The conversion of unmethylated cytosine to uracil by bisulfite sodium provides a completely unmethylated cytosine for MSP. The modified gDNA was stored at −80 °C and used within a week for analysis. Target regions of the mouse FTH1 locus were chosen using MethPrimer (http://www.urogene.org/methprimer accessed on 19 April 2022) on regions upstream (−5 kb) and downstream (+3 kb) of the FTH1 transcription start site. Methylation-specific primers were designed using the METH Primer tool (http://www.urogene.org/cgi-bin/methprimer/methprimer.cgi accessed on 19 April 2022). CpG islands were predicted by the online Meth Primer tool at the default criteria (C+Gs/total bases >50%, CpG observed/expected >0.6).

### 2.11. Prediction of Transcription Factor (TF) Binding Sites by Bioinformatics Analysis

To predict the TF binding sites using the online database, we obtained the FTH promoter sequence from the Ensembl database (http://www.ensembl.org accessed on 19 April 2022) to search for putative TF binding sites of the mouse FTH1 upstream 1.0 kb sequence. The PROMO ALGGEN promoter mapping program (http://alggen.lsi.upc.es/cgi-bin/promo_v3/promo/promoinit.cgi?dirDB=TF_8.3 accessed on 19 April 2022), the TFBIND database (http://tfbind.hgc.jp accessed on 19 April 2022), and TRANCFAC (http://www.gene-regulation.com accessed on 19 April 2022) were used. The program parameters in PROMO ALGEEN were set with a cutoff for the dissimilarity matrix of 15% or less, which was the default setting. The TF matrix from the TRANFAC library was performed with the default setting of matrix similarity of 0.85. Those with a score of 0.85 or higher were selected from the list of TFBIND.

### 2.12. Statistical Analysis

SPSS analyses were performed with the help of IBM SPSS Statistics package version 25 (IBM Corp., Armonk, NY, USA). The results are presented as mean ± SD. Statistical significance was tested using Student’s *t* test among the experimental groups (CTRL, HFe), with *p* values < 0.05 as the criterion for statistical significance.

## 3. Results

### 3.1. High Iron Intake in the Hfe Decreased Body Weights and Fat Mass

Hepatic iron contents measured by the ferrozine assay and plasma ferritin level were higher in HFe group than in CTRL (Figure 1b,c). Prussian blue staining showed elevated iron levels both in the liver and fat in the HFe group compared to the CTRL group (Figure 1d,e). Cumulative food intake did not differ between the groups (Figure 1f). At euthanization, fat mass was found to be significantly lower in the HFe group than in the CTRL group (Figure 1g). The food efficiency ratio (FER) in the HFe group was lower than that in CTRL (Figure 1h). At sacrifice, fat mass was found to be significantly lower in the HFe group than in the CTRL group, while the weights of other organs, such as the heart and liver, were similar between the groups (Figure 1i,j). These data suggest that high iron intake from the gestational age of 8 weeks until 30 weeks after birth resulted in reduced fat mass accumulation and lowered body weight.

### 3.2. Iron-Metabolism Gene Expressions Were Affected by HFe Diet

Hepatic hepcidin, an iron-regulating hormone that plays a major role in systemic iron homeostasis in the liver by iron overload [32], was upregulated in the HFe group compared to that in the CTRL (Figure 2a). TfR1 and/or TfR2, which are receptors for transferrin-bound iron uptake into the cells, were downregulated in the liver, fat, and small intestine of the HFe group (Figure 2a–c). However, there were significant reductions in both mRNA and protein levels of FTH (an iron storage protein) in the liver, fat, and small intestine in the HFe group compared to the CTRL group (Figure 2d). In contrast, there was no significant difference in the FTL protein levels in the liver, fat, and small intestine (Figure 2a–c). Our data indicated that the HFe diet altered the expression of iron metabolism genes with the expected upregulation of hepcidin and TfR genes and an unexpected downregulation of the FTH gene.

### 3.3. Low Gene Expression of FTH in Pups Was Not Associated with DNA Methylation in CpG Island

We speculated hypermethylation in FTH1 as a possible cause of low FTH1 expression in HFe pups, referring to a study by Laker et al. [33]. In the MethFinder database, two putative CpG islands (CGIs) of the mouse FTH1 sequence were detected, one within the coding region (from +1855 to +2475) and the other in the promoter region (from −3447 to −3330) (Figure 3a). To investigate the occurrence of gene hypermethylation in offspring due to the HFe diet, methylation-specific polymerase reaction (MSP) analysis was performed to detect methylated sequences in the liver and adipose tissues. Our results showed that both the CTRL and HFe groups had no detectable methylation products in the putative CGIs in the FTH1 promoter region and in the putative CGIs of the protein-coding region (from +2273 to +4345), suggesting no significant changes in the methylation status due to HFe (Figure 3b,c).

We investigated FTH expression at the transcriptional level by exploring putative transcription factors (TFs) and their expression levels. Using three online databases, we identified 15 putative TFs commonly found in the promoter regions of FTH (Appendix A, Figure 3d). After matching the transcription factor binding sites, 10 out of 15 TFs were identified as putative TFs: C/EBP family, C/EBPα, C/EBPβ, GATA1, GATA2, AP1, SRF, YY1, CP2, and Nrf2 (Figure 3e). Among these, C/EBPα, C/EBPβ, GATA2, and Nrf2 were further tested because of their possible role in lipid and oxidative stress metabolism [34,35,36,37]. The adipose tissue of HFe showed a significant reduction in the protein levels of C/EBPα, a main transcription factor in adipocyte differentiation [34] (Figure 3f), but there were no significant differences in C/EBPβ, which is associated with earlier adipogenesis, regulation of C/EBPα [34], GATA2, suppression of adipocyte differentiation [37] and nuclear factor erythroid 2-related factor 2 (Nrf2), a survival response against oxidative stress [38] (Figure 3g). Taken together, HFe may not modulate the methylation status in the FTH1 gene but may lower the expression of transcription factors such as C/EBPα to downregulate FTH1 expression.

### 3.4. The HFe Decreased the Adipocyte Size and TG Levels in Plasma and Fat Tissues

To identify the effects of HFe on lipid and glucose metabolism, we analyzed serum lipid and glucose profiles, adipocyte size, and lipid content and performed OGTT between HFe and CTRL (Figure 4). The HFe group exhibited decreased plasma TG levels, smaller adipocytes, and diminished TG content in fat tissues compared to CTRL (Figure 4a,b). However, there were no changes in the plasma levels of free fatty acid (FFA), total cholesterol (TCHO), high-density lipoprotein cholesterol (HDL-C), low-density lipoprotein cholesterol (LDL-C), liver TG, and liver and fat TCHOs (Figure 4b–e). With regard to glucose metabolism, the 15 min OGTT in the HFe group was significantly lower than that in the control group, but the area under the curve (AUC) of the OGTT and fasting glucose levels did not differ between the groups (Figure 4f–h). Our data indicate that HFe altered lipid metabolism, resulting in lower TG content and smaller fat tissues without significant changes in glucose metabolism.

### 3.5. HFe Diet Affected Differentiation of Adipocyte, FA Biosynthesis, and FA β-Oxidation

The adipocyte gene expression levels were investigated to further understand the effects of HFe on adipose tissues. In terms of adipocyte differentiation, we found that HFe lowered mRNA levels of PPARγ, a primary transcription factor involved in the differentiation of adipose tissues and elevated mRNAs levels of additional sex combs, such as 1 (Asxl1), an inhibitor of adipogenesis [39] in comparison to CTRL (Figure 5a). However, Pref-1, a suppressor of adipocyte differentiation, was not significantly affected by HFe treatment (Figure 5a). As for adipose de novo lipogenesis and lipid droplet formation, HFe decreased the expression levels of SREBP1c, a transcription factor involved in fatty acid synthesis [40] (Figure 5b,d) along with SREBP1c’s target genes including FASN and SCD1 (Figure 5b), and it lowered the expression levels of perilipin 1, a regulator of lipid droplet size (Figure 5b). In contrast, there was no significant alteration in LXR α/β, the stimulator of SREBP1c [40] (Figure 5b,d). The relative expression of FA metabolism, including uptake and oxidation functions, cluster of differentiation 36 (CD36), and genes associated with FA absorption [41] were found to be downregulated in the HFe group compared to CTRL, whereas lipoprotein lipase (LPL) mRNA levels were unchanged (Figure 5c). Uncoupling protein 2 (UCP2) and adiponectin were upregulated in the HFe group (Figure 5c), but no changes in mRNA levels of carnitine palmitoyltransferase 1a (CPT1a) were seen, and lower levels of PPARα were detected in the HFe group (Figure 5c). There was a significant reduction in the phosphorylation levels of AMP-activated protein kinase (pAMPK), an activated form of AMPK [42]. These data imply that HFe might contribute to a reduction in fat mass by inhibiting adipose differentiation and de novo FA synthesis.

As a result of insulin sensitivity affected by altered adipocyte differentiation [43], IRS1 was relatively higher in the HFe group than in CTRL, whereas no difference was detected in IRS2 levels (Figure 5f). PI3K, activated by tyrosine phosphorylation of IRS-1 and 2 [43], and GSK3β, which regulates PPARγ target gene expression [44], was not significantly different between the two groups. However, HFe decreased GLUT4, an insulin-responsive glucose transporter [43]. Overall, these results suggest that the inhibition of adipose tissue differentiation by HFe may have an impact on glucose transport and absorption.

### 3.6. Prenatal to Postnatal Exposure of HFe Diet Enhanced the Antioxidant System and Induced Depletion of GSH

Considering that excess iron exposure may cause oxidative stress, which has been associated with aberrant lipid metabolism [45], we next investigated whether HFe mice might have experienced oxidative stress along with lipid alterations. There was no significant difference in MDA (Figure 6a), a marker of lipid peroxidation [46], between CTRL and HFe, but the HFe exhibited higher expression levels of GPX4, an anti-lipid peroxidase that reduces lipid hydrogen peroxide (LOOH) in the presence of GSH [47,48] (Figure 6b). In addition, HFe significantly increased antioxidant enzymes, including HO-1, a catalyst for free iron decomposition of heme [49], and NQO1, a reducer of superoxide radicals and other inorganic species, such as iron (III) ions [50,51] but no significant change was observed in catalase (Figure 6b). The total glutathione (tGSH), an antioxidant that scavenges free radicals and nitrogen species [52], was significantly reduced in the HFe compared to the CTRL (Figure 6c), suggesting a higher use of tGSH in HFe than in CTRL. GSH disulfide reductase (GR), glutaredoxin 1 (GRX1), and GSH S-transferase (GST), which are involved in the regulation of GSH synthesis by GSSG and GSH, showed no differences between HFe and CTRL (Figure 6d). Taken together, our data suggest that long-term exposure to HFe enhances the expression of antioxidant enzymes and results in depletion of tGSH.

## 4. Discussion

Our study provides evidence of the effects of high-dose iron supplementation—from the fetal to adult stage in mice—on postnatal growth and lipid metabolism. HFe diet-fed offspring exhibited a smaller body size, decreased body fat mass, and smaller adipocyte size than those without the HFe diet. These effects appeared to be related to a reduction in TG synthesis in adipose tissues.

The nutritional status of the utero may induce long-term metabolomic changes [53,54]. Possible associations between maternal diets such as maternal high-fat diets [55], protein-restricted diets [56], and iron-restricted diets [53,54] during pregnancy and outcomes in the next generation have been reported. Samuelsson et al. reported that maternal obesogenic diets containing high fats and sugars during pregnancy and lactation led to abnormally large adipocytes in the next generation in a mouse model [55], suggesting that lipid metabolism in offspring is affected by fetal nutritional status. Offspring born from mice fed an iron-deprived diet 7 days before mating showed improved glucose tolerance but decreased hemoglobin, red blood cell (RBC) count, hematocrit, RBC volume, and severe anemia compared with control diet-fed mice, accompanied by persistent lower body weight and TG level until 3 months of age [54]. In a study by Lisle et al., rats born to mothers fed an Fe-deficient diet during pregnancy exhibited significant weight loss, growth retardation, and hypertension in later life compared to controls [53]. To the best of our knowledge, there are little data on the consequences of excess iron exposure due to maternal diets from the fetal stage to adulthood. A clinical study demonstrated that extra iron accumulation in pregnancy, even as early as the first trimester, may increase the risk of gestational diabetes mellitus [23]. In addition, exposure to excessive iron supplementation during gestation leads to an increased prevalence of type 1 diabetes in offspring [57]. Recently, Guo et al. reported that both high (344 mg/kg) and low (2 mg/kg) iron supplementation during gestation was associated with abnormal placental iron transport and gland development in fetuses [58].

Our investigation on the long-term effects of maternal iron excess on offspring may provide evidence to determine optimal iron supplementation levels during pregnancy to minimize damage to maternal and fetal health. First, a high maternal iron diet during pregnancy led to high iron accumulation in the liver tissues of the offspring but less clearly in the adipose tissues. We observed that high hepatic iron levels in the offspring were accompanied by elevated hepcidin expression levels and low levels of TfR1 and 2, which were similar to those exposed to high dietary iron during adulthood [32]. Maternal iron is taken up in the transferrin bound form through the placenta into the fetus [59]. Therefore, extreme maternal iron overload may impair placental iron intake following iron overload in the fetus [59]. Despite the high hepatic iron levels, we found no elevation in FTH1 mRNA levels in the offspring. FTH1 is known to be elevated in response to high iron levels as a cellular iron storage protein and major regulator of the cellular labile iron pool [60]. Iron supplementation in both iron-deficient anemia mice and their offspring significantly upregulated FTL levels with improved physical growth compared to untreated mice [61]. We have seen low FTH1 and increased iron accumulation, with adipocyte size reduced by 45.5%, adipose tissue TG decreased by 60%, and serum TG decreased by 56.6%; however, the interaction between them is yet to be studied. Iron-enriched diet containing 3% carbonyl iron for 16 weeks led to statistically decreased mean adipocyte size by approximately 36% in visceral adipose tissue of mice [62]. Previously, the partial influence of both fetal and maternal ferritin levels on the birth weight of newborns was reported in a mouse model [63]. Furthermore, we considered the possible interaction between the identified decline in FTH levels and lipid metabolism in the HFe offspring.

Lipid metabolism in the offspring is known to be sensitive to abnormal maternal nutritional status in several studies [64]. A copper-deficient diet (0.5 mg/kg diet) during the fetal stage and followed by weaning led to C57BL mice exhibiting reduced lipid content in organs such as the liver, heart, and kidneys [64]. The offspring of mother mice given a magnesium deficient diet (50 mg/kg elemental diet) showed reduced growth and survival with a lower monounsaturated fatty acid (MUFA) [65]. Maternal iron-deficient diets also affect lipid metabolism [66,67]. Mothers and newborn rats that received an iron-deficient diet (5 ppm) had hyperlipidemia due to elevated serum TGs, cholesterol, and phospholipids compared to rats fed the control diet (307 ppm) [66]. When guinea pigs were fed an iron-deficient diet from mating to lactation, their grown-up pups revealed alterations in essential fatty acid metabolism, such as higher dihomogammalinolenic acid, docosapentaenoic acid, and DHA contents in the brain [67]. Prenatal iron restriction caused newborn mice to have suppressed lipogenesis and bile acid synthesis, as evidenced by the observation of decreased hepatic TG concentrations with downregulated levels of SREBP1c, LXRα, and CD36 [68]. Meanwhile, to the best of our knowledge, few studies have investigated the effects of excessive iron in the maternal diet on lipid metabolism in the next generation. Elevated levels of lipid peroxidation indicators, including pentane and ethane, have been reported in animals injected with iron-dextran (460 mg of iron per 100 g body weight) [69]. In an adult rat model, hepatic iron overload induced lipid peroxidation with 3% iron carbonyl diet for 12 weeks, which was accompanied by increased plasma TG and cholesterol and decreased hepatic activity of HMG-CoA and cholesterol 7α-hydroxylase [70]. Here, we demonstrated that prenatal perturbations in iron metabolism caused by a high-iron diet led to decreased adipocyte size and TG levels in the offspring.

To understand the molecular mechanisms underlying the lipid metabolic effects of the pre- and postnatal HFe diet in mice, gene expression levels in adipocyte differentiation, lipogenesis, and FA oxidation were investigated. Considering that PPARγ and C/EBPα are known to be key factors in adipocyte differentiation [71], their lower expression levels in Hfe-fed mice may have contributed to the lower adipocyte diameter. According to the correlation analysis between hepatic iron content and gene transcription performed with unpublished data, genes including Asxl1, SCD1, Acsl5, leptin, and GATA were significantly correlated with iron levels (Appendix A). Among these genes, Asxl1, which was significantly increased in the HFe group in our study, has been demonstrated to suppress PPARγ and its related genes by binding to the heterochromatin protein 1 binding domain [39]. Accordingly, the HFe-induced decrease in PPARγ and C/EBPα in our study might be associated with increased mRNA levels of Asxl1. In addition, we observed a decrease in SREBP1c and its target genes, including FASN and SCD1, leading to reduced TG levels in adipocytes and hindered lipid accumulation. SREBP1c is an insulin-stimulated FA synthesis-stimulating transcription factor that increases the transcription of genes encoding ACC1, FASN, and SCD1 and that regulates plasma TG levels [72]. Taken together, alteration in the expression level of Asxl1 due to HFe inhibits adipocyte differentiation through PPARγ, and it might have contributed to suppression of SREBP1c in de novo FA synthesis and reduction of fat accumulation in long exposure of HFe from birth in mice.

A link between excess iron and oxidative stress necessitates verification of the mechanism linking iron and lipid oxidation. An iron-rich diet in mice increases hepatic lipid oxidation and upregulates PPARα [73]. In contrast, Bonomo et al. [74] reported downregulation of PPARα in an iron-injected rodent model, which led to elevated serum cholesterol levels. We observed downregulation in the PPARα expression levels and its target gene, CD36, and activation of pAMPK in the offspring with pre- and postnatal high iron exposure. As ROS regulates PPAR family receptors [20], we may assume that the decreased PPARα might have been influenced by HFe-produced ROS. CD36 is associated with the transport and absorption of FA [75] and is responsive to ROS levels [76]. As reported earlier, dietary iron overload in a mouse model effectively inhibited the upregulation of CD36 induced by a high-fat diet [77], which may support our results of reduced CD36 levels in HFe group. In addition, AMPK is associated with a reduction in body fat content by increasing lipid peroxidation in the liver and tissues [78]. Specifically, AMPK effectively inhibited excess iron-generated ROS in vivo [79] and in vitro [80]. AMPK activation is associated with the elevation of PPARα activation and reduction in FA oxidation [81]. Concurrently, the decreased pAMPK and PPARα levels seen in our study, which in turn decreased lipid oxidation [82], might be explained as compensation for FA loss due to HFe.

Adiponectin increases with excess iron injection in vitro, leading to the activation of hepatic FA oxidation [83] and synthesis [84]. Moreover, adiponectin is associated with the upregulation of UCP2 by stimulating mitochondrial superoxide production [85]. Similarly, our data revealed higher adiponectin and UCP2 levels in HFe. Adiponectin levels increase with hereditary iron overload in a mouse model [86]. UCP2 participates in dietic thermogenesis [87]. Considering the roles of adiponectin and UCP2, our findings might be associated with HFe-elevated adiponin-induced lipid oxidation and upregulated UCP2, which consequently reduces fat via thermogenesis.

Elevated oxidative stress is an established mechanism that results from iron-induced cellular dysfunction [84]. We identified decreased tGSH levels in the HFe group, although we found no significant differences in MDA and lipid peroxidation products between the HFe and CTRL groups. A study by Aydın et al. [30] revealed that iron overload leads to a decrease in the tGSH and glutathione/oxidized glutathione ratio. Oxidative stress was suggested to first induce GSH depletion and then increase ROS in tissues [88], which is considered the result of the actions of increased antioxidant products to prevent ROS generation or to remove them [89]. We also found that HFe-induced elevation of antioxidant gene products, including HO-1, GPX4, and NQO1, participates in iron redox conversion [90,91]. In addition, GSH-depleted mice had lower hepatic TG concentrations and lower mRNA levels of SREBP1c, DGAT2, and FASN, suggesting that increased ROS levels due to GSH deficiency suppressed adipogenesis [92]. Thus, HFe-elevated antioxidant gene expression and reduced tGSH levels might have reduced oxidative stress and altered lipid metabolism.

The cross-regulatory loop between PPARγ and C/EBPα during adipose differentiation is known to affect glucose sensitivity by regulating IRS1 transcription [43] and translocation of GLUT4, an insulin-responsive glucose transporter [93]. We detected that HFe increased IRS1 mRNA with decreased mRNA levels of GLUT4, in addition to low expression levels of PPARγ and C/EBPα. C/EBPα (-/-) cells exhibited PPARγ-stimulated expression of GLUT4 but less glucose uptake than C/EBPα wild-type cells [43]. Previously, C/EBP binding sites were found to be present in the promoters of IR/IRS-1 and GLUT4 [94,95]. It can be hypothesized that our data on HFe-induced changes after 15 min of OGTT might be due to HFe-altered mRNA levels of IRS1 and GLUT4, which are regulated by C/EBPα during adipogenic differentiation. Collectively, these results indicate that HFe might play a main role in the aberrant lipid metabolism observed in our study, possibly suggesting the potent existence of glucose uptake and sensitivity with some effects of weight and body fat loss.

The observation of the offspring’s low levels of FTH led us to investigate possible modifications in the methylation status of the promoter regions of the FTH gene. Long-term intake of some nutrients has been shown to regulate the methylation status of genes [96,97]. A high-fat diet (HFD) for 16 weeks reduced hepatic TfR2 expression and decreased hepatic iron content due to DNA hypermethylation of the promoter and suppressed expression of the transcription factor hepatocyte nuclear factor 4α [96]. In addition, folic acid-supplemented diets during pregnancy and lactation in both mothers and newborns significantly alter the methylation of PPARγ, ERα, p53, and APC [97]. However, in our study, a high-iron diet during pre- and postnatal periods did not induce changes in the methylation status of two putative FTH1 CpG islands (from +1855 to +2475, from −3447 to −3330). Other epigenetic processes, including histone modification and genomic imprinting, may also alter the pattern and activity of gene expression [98] and may occur independently of gene silencing [99]. HF diet exposure in utero increased the prevalence of metabolic syndromes-like phenomena through histone methylation of adipocytokine, adiponectin, and leptin gene expression in the adipose tissue of offspring mice [100]. The gestational HF diet induced upregulation of Pck1 by specific histone modifications and transcriptional activation in the coding and upstream regions of the gene [101]. As we found no relationship between DNA methylation and aberrant FTH expression, there might have been other epigenetic mechanisms, such as histone modification, for the dysregulated expression of FTH.

To further elucidate the mechanism underlying FTH1 downregulation, we identified potential TFs and binding sites in the FTH promoter region. We used several online bioinformatic resources [102]. Singha et al. identified putative TF-binding sites using various online databases to contribute to anticancer therapy through the regulation of chemokines in ovarian cancer cells and tissues [103]. We identified 10 candidate TFs (C/EBP family, C/EBPα, C/EBPβ, GATA1, GATA2, AP1, SRF, YY1, CP2, and Nrf2) using online databases including PROMO ALGGEN, TF BIND, and TRANSFAC. Some TFs, including GATA1, GATA2, YY1, and C/EBPα, were referred to in other investigations on genes relevant to iron metabolism regulation or expression of FTH [104,105,106]. GATA1 cooperates with bioavailable iron to synthesize heme in red blood cells, which in turn regulates red blood cell differentiation by downregulating target and mitotic spindle genes [104]. GATA2 is a key factor in maintaining hepcidin iron homeostasis, as EMSA results revealed GATA2′s involvement in the promoter region of hepatic *hamp1* in mice [105]. YY1, induced by the overexpression of B-cell translocation gene 2, is involved in hepatic hepcidin production by regulating iron homeostasis, as it is recruited to the promoter of hepcidin to significantly elevate serum hepcidin levels [106]. Among the 10 putative TFs in the mouse FTH1 promoter region, we only detected changes in the gene expression levels of C/EBPα, with reduced protein levels in the HFe group. C/EBPα protein has been suggested to be one of the key factors in hepcidin-regulated iron homeostasis [107]. Courselaud et al. discovered that C/EBPα-deleted mice exhibited hepatic iron overload with decreased hepcidin mRNA levels accompanied by iron accumulation in periportal hepatocytes [107]. Our data hint at the interaction between decreased C/EBPα protein and low expression of FTH in iron overload status without a change in FTL. Therefore, the Hfe diet in the fetus may have contributed to the low expression of FTH by downregulating C/EBPα, leading to abnormal lipid metabolism.

However, the main limitation of our study was that we did not consider the association between iron and other minerals. Iron absorption has been reported to suppress several other trace minerals, such as zinc and copper [108]. Iron supplementation in the third trimester of pregnancy in Peru significantly reduced the absorption of zinc [109], indicating that the transfer of zinc between the mother and fetus may be influenced by iron supplementation, leading to adverse effects. Therefore, further studies are recommended to consider the link between iron and other mineral metabolism and to investigate the possibility of other adverse effects.

In conclusion, mice exposed to HFe from the fetal stage to adulthood showed decreased fat accumulation and dysregulated iron metabolism, with lower FTH1 expression. Although the exact mechanism by which FTH1 regulates iron homeostasis in mice requires further investigation, alterations in HFe-induced lipid metabolism might have negatively affected the growth of their offspring. The suggested mechanism by which maternal HFe influences the offspring is outlined in Figure 7, although further investigations are required to determine the effects of maternal iron status on fetal lipid metabolism. Our data provide additional evidence to support the establishment of guidelines for iron supplementation in pregnant women.

## Figures and Tables

**Figure 1 nutrients-14-02451-f001:**
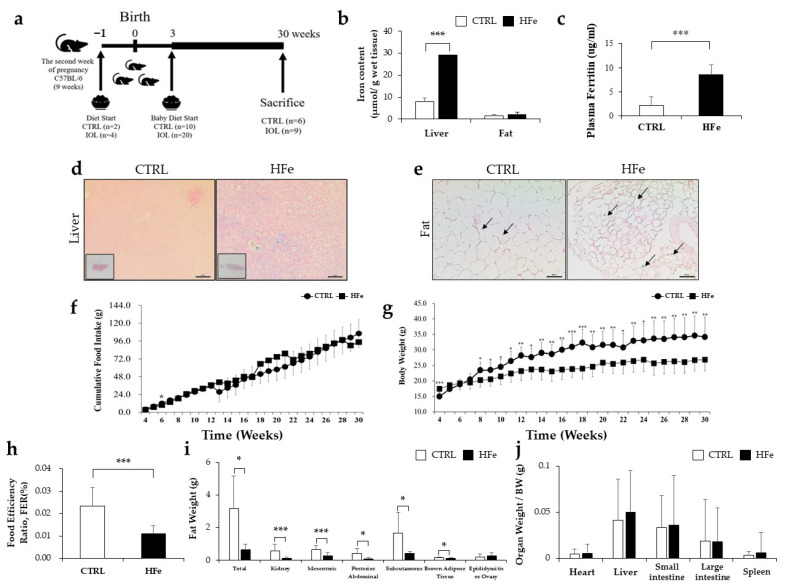
Excessive dietary iron intake decreased body weight and fat mass. Diagram for feeding schedule (**a**) and amount of iron in liver and fat as measured by (**b**) ferrozine assay, plasma ferritin level (**c**), Prussian Blue Staining in the (**d**) liver and (**e**) mesenteric fat. (**f**) Cumulative food intake, (**g**) body weight, (**h**) food efficiency ratio (FER), (**i**) fat weight, and (**j**) organ weight at final week (week 30) of each group (CTRL, HFe). Data are expressed as means  ±  SE; Student’s t test was used for the test of difference. * *p* < 0.05, ** *p* < 0.01, *** *p* < 0.001. All sections were paraffin embedded. Scale bar, 100 µm; magnification, ×10. Black arrows show blue stain indicating iron accumulation in each tissue.

**Figure 2 nutrients-14-02451-f002:**
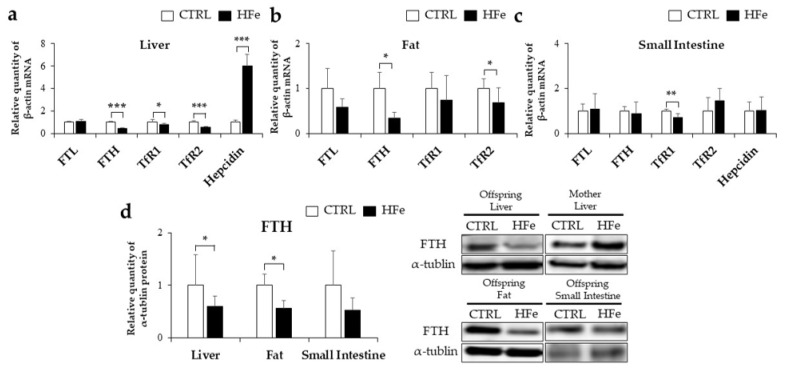
Iron-related mRNA and protein expression was affected by HFe diet. Amount of mRNA in the (**a**) liver, (**b**) fat, and (**c**) small intestine as measured by RT-PCR are presented. (**d**) Effects of iron overload on FTH protein expression levels of mice pups. Results are expressed as mean  ±  SE; Student’s t test was used for the test of difference. * *p <* 0.05, ** *p* < 0.01, *** *p* < 0.001.

**Figure 3 nutrients-14-02451-f003:**
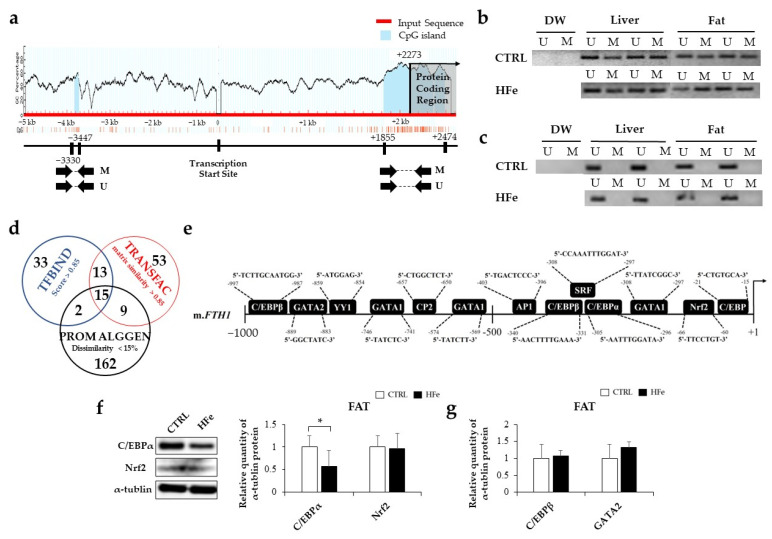
The change in FTH expression was not affected by DNA methylation in CGIs. (**a**) Schematic showing the tested regions for MSP at the FTH1 locus. Target regions were chosen using MethFinder (http://www.urogene.org/methprimer accessed on 19 April 2022) on regions upstream (−5 kb) and downstream from the FTH1 transcription start site. Agarose gel electrophoresis of MSP products are shown on (**b**) regions upstream (−5 kb) and (**c**) downstream (+3 kb) from CTRL- and HFe-fed offspring TFs in the FTH1 gene promoter region and were predicted using (**d**) PROMO ALGGEN, TRANSFAC and TFBIND software programs. (**e**) The binding sites of the TF detected in 3 databases. The spacing of the sites in the diagram reflects the typical spacing of the actual sequence. Solid arrows indicate transcription start sites. (**f**) Amount of protein levels of C/EBPα and Nrf2 and (**g**) mRNA levels of C/EBPβ and GATA2 in fat are presented. Data are expressed as means  ±  SE; Student’s *t* test was used to test differences between means. * *p* < 0.05. M, product amplified with methylated-specific primer; U, product amplified with unmethylated-specific primer; DW, double distilled water control; AP1, activating protein-1; GATA1, GATA-binding protein 1; SRF, serum response factor; YY1, Yin Yang 1.

**Figure 4 nutrients-14-02451-f004:**
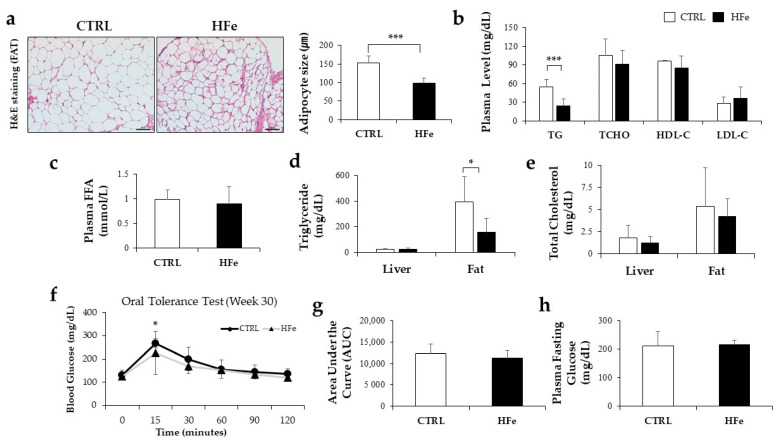
HFe decreased the adipocyte size and TG levels in the blood and fat tissues. Representative images of hematoxylin and eosin (H&E) staining. (**a**) Decreased adipocyte size in HFe mice. (**b**) Plasma biochemistry related to lipid profiles and (**c**) FFA were measured at week 30. (**d**) TG and (**e**) TCHO levels in liver and fat of each group. (**f**) The results of OGTT, (**g**) AUC of OGTT, and (**h**) plasma fasting glucose at week 30 in mice fed CTRL and HFe diet. Data are expressed as means  ±  SE; Student’s t test was used to test differences between means. * *p* < 0.05, *** *p* < 0.001. TG, triglycerides; TCHO, total cholesterol; HDL-C, high-density lipoprotein cholesterol; LDL-C, low-density lipoprotein cholesterol; FFA, free fatty acids.

**Figure 5 nutrients-14-02451-f005:**
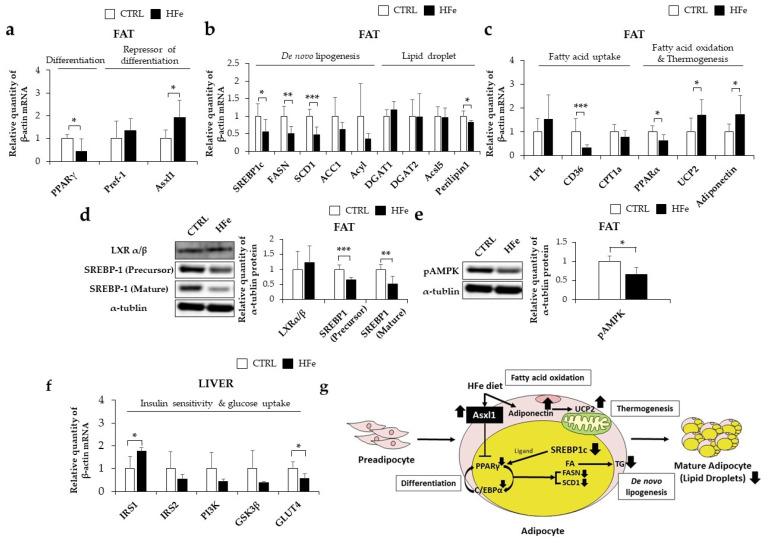
Ann excessive iron diet affected FA biosynthesis and differentiation of adipocytes more than FA β-oxidation. (**a**) Amount of mRNA level in differentiation of adipocyte (**b**) de novo lipogenesis, lipid droplet FA uptake, and FA oxidation (**c**) and in mesenteric fat as measured by RT-PCR. Effects of HFe on protein levels in mesenteric fat related to (**d**) FA synthesis and (**e**) FA oxidation. (**f**) Hepatic mRNA levels of insulin sensitivity and glucose uptake. (**g**) Proposed schematic mechanism of the alteration on lipid metabolism by HFe. Data are expressed as means  ±  SE; Student’s *t* test was used to test differences between means. * *p* < 0.05, ** *p* < 0.01, *** *p* < 0.001.

**Figure 6 nutrients-14-02451-f006:**
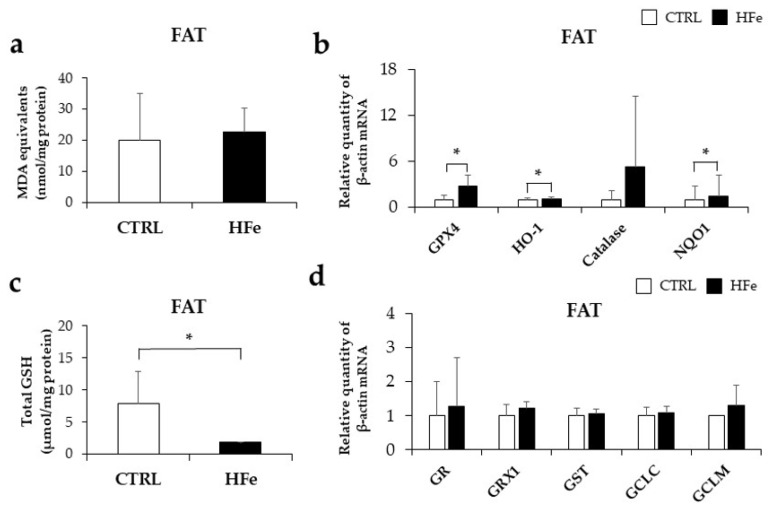
The HFe enhanced Nrf2-related antioxidant expression. (**a**) Estimation of MDA in mesenteric fat of each group (CTRL, HFe). (**b**) mRNA expression related to antioxidant enzymes. (**c**) Contents of tGSH and (**d**) RT-PCR analysis were performed to detect mRNA expression of GSH-Trx and production system in mice. All data are expressed as the mean ± S.E. Student’s *t* test was used to test differences between means. * *p* < 0.05. GPX4, glutathione peroxidase 4; HO-1, heme oxygenase-1; NQO1, quinone oxidoreductase 1; GR, glutathione disulfide reductase; GRX1, glutaredoxin 1; GST, glutathione S-transferase; GCLC, glutamate cysteine ligase catalytic; GCLM, glutamate cysteine ligase modifier.

**Figure 7 nutrients-14-02451-f007:**
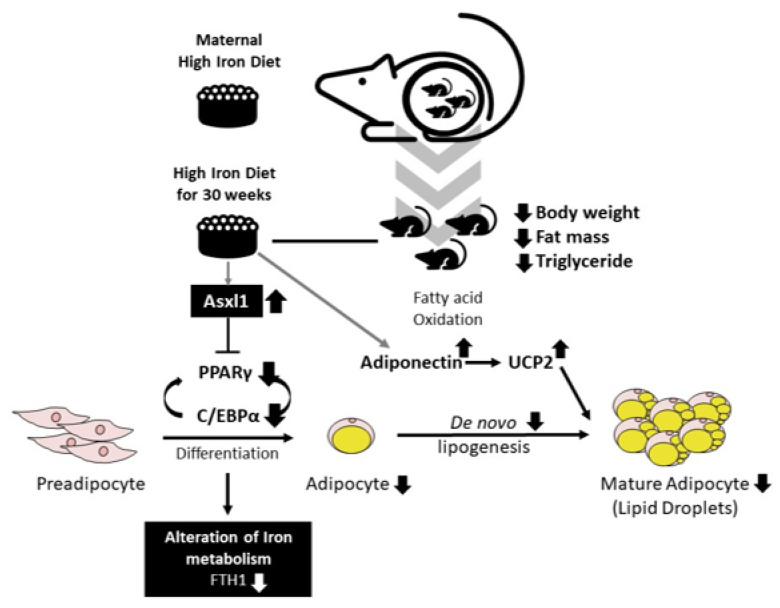
Summarized scheme of the Heft diet affecting iron and lipid metabolism in mice in the present study.

**Table 1 nutrients-14-02451-t001:** Composition of different iron-concentrated diets.

Ingredient (g/1000 g Diet)	AIN-76A Diet	
	CTRL	HFe
Casein	200	200
Corn starch	150	150
Sucrose	499.99	490.09
Corn oil	50	50
Cellulose	50	50
Vitamin mixture	10	10
AIN 76a mineral mix	35	35
Choline bitartrate	2	2
DL-Methionine	3	3
Butylated hydroxytoluene	0.01	0.01
FeSO_4_, 7H_2_O	-	9.9
Total (g)	1000	1000

CTRL: control diet, HFe: High iron diet, FeSO_4_, 7H_2_O: Iron(II) sulfate heptahydrate.

## Data Availability

The dataset supporting the conclusions of this article is included within the article.

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
