# Peer review of "High Iron Exposure from the Fetal Stage to Adulthood in Mice Alters Lipid Metabolism"

_nutrients, 2022, doi:10.3390/nu14122451_

Round 1
Reviewer 1 Report
- In Line 95, they received the pregnant mice at two weeks of gestation (E14). But line 99, they started feeding from 2 weeks before giving birth. This doesn’t make sense because total gestation period is 3 weeks. Please confirm feeding condition and make a diagram for feeding schedule.
- Author did oral glucose tolerance test for mice in line 119. Please give some references on this oral glucose administration. For rodents, many papers reported glucose treatment via intra-peritoneal or tail vein.
- on Figure 1b & 1c, the quality of these figures is not good for publication. So please replace these figures to improved images.
- In figure 2, units for mRNA level or protein level were missing. Please add ‘arbitrary units’ for Y axis. Same issues are in fig3, 5 and 6.
- In figure 2, only Figure 2C showed CTRL with blank square and HFe with filled square. Please give these symbols to other figures (b, d and e) including figure 4, 5 and 6.
- In figure 2, graph for protein on Mother liver is missing. Please confirm it.
- In Line 61, change to ‘When young women without anemia (ferritin level <= 20 ug/L) were supplemented with 240 mg of iron for more than two days, fractional absorption of iron decreased by more than 45%... ‘
- In Line 228, change to ‘At euthanization, fat mass was found to be significantly lower in the HFe group than in the CTRL group….
- In Line 427, change to ‘We observed that high hepatic iron levels in the offspring were accompanied by elevated hepcidin expression levels and low levels of TfR1 and 2, which were similar to those exposed to high dietary iron during adulthood.
- In Line 448, change to ‘The offspring of mother mice given a magnesium deficient diet (50 mg/kg elemental diet) showed reduced growth and survival with a lower monounsaturated fatty acid (MUFA).
Reviewer 2 Report
The authors investigate long-term iron supplementation on prenatally treated mice which continiued to adulthood of the offspring. The authors should clarify the experimental set up as at this point it raises certain questions as:
1. Was daily food intake monitored?
2. Were pregnant mice housed in individual cages? Was the offspring housed in individual cages following weaning? How did the authors ensure that all experimental animals obtained the same amount of Fe?
3. Did the two groups contain both male and female mice? Why do the authors cite the Declaration of Helsinki if their experiment is on mice?
4. Did the authors observe any hematological changes?
5. If liver organ weight was the same in both control and Fe-treated mice, do the authors have data regarding spleen weight as it is a site for extramedullary hematopoiesis?
6. Did the authors observe signs of ferroptosis?
7. References should be reduced
